# Representer Point Selection for Explaining Deep Neural Networks

**Chih-Kuan Yeh**[*]    **Joon Sik Kim** [*]    **Ian E.H. Yen**    **Pradeep Ravikumar**
Machine Learning Department
Carnegie Mellon University
Pittsburgh, PA 15213
{cjyeh, joonsikk, eyan, pradeepr}@cs.cmu.edu

## Abstract

We propose to explain the predictions of a deep neural network, by pointing to the set of what we call *representer points* in the training set, for a given test point prediction. Specifically, we show that we can decompose the pre-activation prediction of a neural network into a linear combination of activations of training points, with the weights corresponding to what we call representer values, which thus capture the importance of that training point on the learned parameters of the network. But it provides a deeper understanding of the network than simply training point influence: with positive representer values corresponding to excitatory training points, and negative values corresponding to inhibitory points, which as we show provides considerably more insight. Our method is also much more scalable, allowing for real-time feedback in a manner not feasible with influence functions.

## 1 Introduction

As machine learning systems start to be more widely used, we are starting to care not just about the accuracy and speed of the predictions, but also *why* it made its specific predictions. While we need not always care about the why of a complex system in order to trust it, especially if we observe that the system has high accuracy, such trust typically hinges on the belief that some other expert has a richer understanding of the system. For instance, while we might not know exactly how planes fly in the air, we trust some experts do. In the case of machine learning models however, even machine learning experts do not have a clear understanding of why say a deep neural network makes a particular prediction. Our work proposes to address this gap by focusing on improving the understanding of experts, in addition to lay users. In particular, expert users could then use these explanations to further fine-tune the system (e.g. dataset/model debugging), as well as suggest different approaches for model training, so that it achieves a better performance.

Our key approach to do so is via a representer theorem for deep neural networks, which might be of independent interest even outside the context of explainable ML. We show that we can decompose the pre-activation prediction values into a linear combination of training point activations, with the weights corresponding to what we call representer values, which can be used to measure the importance of each training point has on the learned parameter of the model. Using these representer values, we select representer points – training points that have large/small representer values – that could aid the understanding of the model's prediction.

Such representer points provide a richer understanding of the deep neural network than other approaches that provide influential training points, in part because of the meta-explanation underlying our explanation: a positive representer value indicates that a similarity to that training point is *excita-*

---

[*]Equal contribution

*tory*, while a negative representer value indicates that a similarity to that training point is *inhibitory*, to the prediction at the given test point. It is in these inhibitory training points where our approach provides considerably more insight compared to other approaches: specifically, what would cause the model to *not* make a particular prediction? In one of our examples, we see that the model makes an error in labeling an antelope as a deer. Looking at its most inhibitory training points, we see that the dataset is rife with training images where there are antelopes in the image, but also some other animals, and the image is labeled with the other animal. These thus contribute to inhibitory effects of small antelopes with other big objects: an insight that as machine learning experts, we found deeply useful, and which is difficult to obtain via other explanatory approaches. We demonstrate the utility of our class of *representer point* explanations through a range of theoretical and empirical investigations.

## 2   Related Work

There are two main classes of approaches to explain the prediction of a model. The first class of approaches point to important input features. Ribeiro et al. [1] provide such feature-based explanations that are model-agnostic; explaining the decision locally around a test instance by fitting a local linear model in the region. Ribeiro et al. [2] introduce Anchors, which are locally sufficient conditions of features that "holds down" the prediction so that it does not change in a local neighborhood. Such feature based explanations are particularly natural in computer vision tasks, since it enables visualizing the regions of the input pixel space that causes the classifier to make certain predictions. There are numerous works along this line, particularly focusing on gradient-based methods that provide saliency maps in the pixel space [3, 4, 5, 6].

The second class of approaches are sample-based, and they identify training samples that have the most influence on the model's prediction on a test point. Among model-agnostic sample-based explanations are prototype selection methods [7, 8] that provide a set of "representative" samples chosen from the data set. Kim et al. [9] provide criticism alongside prototypes to explain what are not captured by prototypes. Usually such prototype and criticism selection is model-agnostic and used to accelerate the training for classifications. Model-aware sample-based explanation identify influential training samples which are the most helpful for reducing the objective loss or making the prediction. Recently, Koh and Liang [10] provide tractable approximations of influence functions that characterize the influence of each sample in terms of change in the loss. Anirudh et al. [11] propose a generic approach to influential sample selection via a graph constructed using the samples.

Our approach is based on a representer theorem for deep neural network predictions. Representer theorems [12] in machine learning contexts have focused on non-parametric regression, specifically in reproducing kernel Hilbert spaces (RKHS), and which loosely state that under certain conditions the minimizer of a loss functional over a RKHS can be expressed as a linear combination of kernel evaluations at training points. There have been recent efforts at leveraging such insights to compositional contexts [13, 14], though these largely focus on connections to non-parametric estimation. Bohn et al. [13] extend the representer theorem to compositions of kernels, while Unser [14] draws connections between deep neural networks to such deep kernel estimation, specifically deep spline estimation. In our work, we consider the much simpler problem of explaining pre-activation neural network predictions in terms of activations of training points, which while less illuminating from a non-parametric estimation standpoint, is arguably much more explanatory, and useful from an explainable ML standpoint.

## 3   Representer Point Framework

Consider a classification problem, of learning a mapping from an input space $\mathcal{X} \subseteq \mathbb{R}^d$ (e.g., images) to an output space $\mathcal{Y} \subseteq \mathbb{R}$ (e.g., labels), given training points $\mathbf{x}_1, \mathbf{x}_2, ...\mathbf{x}_n$, and corresponding labels $\mathbf{y}_1, \mathbf{y}_2, ...\mathbf{y}_n$. We consider a neural network as our prediction model, which takes the form $\hat{\mathbf{y}}_i = \sigma(\Phi(\mathbf{x}_i, \boldsymbol{\Theta})) \subseteq \mathbb{R}^c$, where $\Phi(\mathbf{x}_i, \boldsymbol{\Theta}) = \boldsymbol{\Theta}_1 \mathbf{f}_i \subseteq \mathbb{R}^c$ and $\mathbf{f}_i = \Phi_2(\mathbf{x}_i, \boldsymbol{\Theta}_2) \subseteq \mathbb{R}^f$ is the last intermediate layer feature in the neural network for input $\mathbf{x}_i$. Note that $c$ is the number of classes, $f$ is the dimension of the feature, $\boldsymbol{\Theta}_1$ is a matrix $\subseteq \mathbb{R}^{c \times f}$, and $\boldsymbol{\Theta}_2$ is all the parameters to generate the last intermediate layer from the input $\mathbf{x}_i$. Thus $\boldsymbol{\Theta} = \{\boldsymbol{\Theta}_1, \boldsymbol{\Theta}_2\}$ are all the parameters of our neural network model. The parameterization above connotes splitting of the model as a feature model $\Phi_2(\mathbf{x}_i, \boldsymbol{\Theta}_2)$ and a prediction network with parameters $\boldsymbol{\Theta}_1$. Note that the feature model $\Phi_2(\mathbf{x}_i, \boldsymbol{\Theta}_2)$

can be arbitrarily deep, or simply the identity function, so our setup above is applicable to general feed-forward networks.

Our goal is to understand to what extent does one particular training point $\mathbf{x}_i$ affect the prediction $\hat{\mathbf{y}}_t$ of a test point $\mathbf{x}_t$ as well as the learned weight parameter $\boldsymbol{\Theta}$. Let $L(\mathbf{x}, \mathbf{y}, \boldsymbol{\Theta})$ be the loss, and $\frac{1}{n}\sum_i^n L(\mathbf{x}_i, \mathbf{y}_i, \boldsymbol{\Theta})$ be the empirical risk. To indicate the form of a representer theorem, suppose we solve for the optimal parameters $\boldsymbol{\Theta}^* = \arg\min_{\boldsymbol{\Theta}} \left\{ \frac{1}{n}\sum_i^n L(\mathbf{x}_i, \mathbf{y}_i, \boldsymbol{\Theta}) + g(||\boldsymbol{\Theta}||) \right\}$ for some non-decreasing $g$. We would then like our pre-activation predictions $\Phi(\mathbf{x}_t, \boldsymbol{\Theta})$ to have the decomposition: $\Phi(\mathbf{x}_t, \boldsymbol{\Theta}^*) = \sum_i^n \alpha_i k(\mathbf{x}_t, \mathbf{x}_i)$. Given such a representer theorem, $\alpha_i k(\mathbf{x}_t, \mathbf{x}_i)$ can be seen as the contribution of the training data $\mathbf{x}_i$ on the testing prediction $\Phi(\mathbf{x}_t, \boldsymbol{\Theta})$. However, such representer theorems have only been developed for non-parametric predictors, specifically where $\Phi$ lies in a reproducing kernel Hilbert space. Moreover, unlike the typical RKHS setting, finding a global minimum for the empirical risk of a deep network is difficult, if not impossible, to obtain. In the following, we provide a representer theorem that addresses these two points: it holds for deep neural networks, and for any stationary point solution.

**Theorem 3.1.** *Let us denote the neural network prediction function by $\hat{\mathbf{y}}_i = \sigma(\Phi(\mathbf{x}_i, \boldsymbol{\Theta}))$, where $\Phi(\mathbf{x}_i, \boldsymbol{\Theta}) = \boldsymbol{\Theta}_1 \mathbf{f}_i$ and $\mathbf{f}_i = \Phi_2(\mathbf{x}_i, \boldsymbol{\Theta}_2)$. Suppose $\boldsymbol{\Theta}^*$ is a stationary point of the optimization problem:* $\arg\min_{\boldsymbol{\Theta}} \left\{ \frac{1}{n}\sum_i^n L(\mathbf{x}_i, \mathbf{y}_i, \boldsymbol{\Theta})) + g(||\boldsymbol{\Theta}_1||) \right\}$, *where $g(||\boldsymbol{\Theta}_1||) = \lambda||\boldsymbol{\Theta}_1||^2$ for some $\lambda > 0$. Then we have the decomposition:*

$$\Phi(\mathbf{x}_t, \boldsymbol{\Theta}^*) = \sum_i^n k(\mathbf{x}_t, \mathbf{x}_i, \alpha_i),$$

*where $\alpha_i = \frac{1}{-2\lambda n}\frac{\partial L(\mathbf{x}_i, \mathbf{y}_i, \boldsymbol{\Theta})}{\partial \Phi(\mathbf{x}_i, \boldsymbol{\Theta})}$ and $k(\mathbf{x}_t, \mathbf{x}_i, \alpha_i) = \alpha_i \mathbf{f}_i^T \mathbf{f}_t$, which we call a representer value for $\mathbf{x}_i$ given $\mathbf{x}_t$.*

*Proof.* Note that for any stationary point, the gradient of the loss with respect to $\boldsymbol{\Theta}_1$ is equal to 0. We therefore have

$$\frac{1}{n}\sum_{i=1}^n \frac{\partial L(\mathbf{x}_i, \mathbf{y}_i, \boldsymbol{\Theta})}{\partial \boldsymbol{\Theta}_1} + 2\lambda\boldsymbol{\Theta}_1^* = 0 \quad \Rightarrow \quad \boldsymbol{\Theta}_1^* = -\frac{1}{2\lambda n}\sum_{i=1}^n \frac{\partial L(\mathbf{x}_i, \mathbf{y}_i, \boldsymbol{\Theta})}{\partial \boldsymbol{\Theta}_1} = \sum_{i=1}^n \alpha_i \mathbf{f}_i^T \quad (1)$$

where $\alpha_i = -\frac{1}{2\lambda n}\frac{\partial L(\mathbf{x}_i, \mathbf{y}_i, \boldsymbol{\Theta})}{\partial \Phi(\mathbf{x}_i, \boldsymbol{\Theta})}$ by the chain rule. We thus have that

$$\Phi(\mathbf{x}_t, \boldsymbol{\Theta}^*) = \boldsymbol{\Theta}_1^* \mathbf{f}_t = \sum_{i=1}^n k(\mathbf{x}_t, \mathbf{x}_i, \alpha_i), \quad (2)$$

where $k(\mathbf{x}_t, \mathbf{x}_i, \alpha_i) = \alpha_i \mathbf{f}_i^T \mathbf{f}_t$ by simply plugging in the expression (1) into (2). $\square$

We note that $\alpha_i$ can be seen as the resistance for training example feature $\mathbf{f}_i$ towards minimizing the norm of the weight matrix $\boldsymbol{\Theta}_1$. Therefore, $\alpha_i$ can be used to evaluate the importance of the training data $\mathbf{x}_i$ have on $\boldsymbol{\Theta}_1$. Note that for any class $j$, $\Phi(\mathbf{x}_t, \boldsymbol{\Theta}^*)_j = \boldsymbol{\Theta}_{1j}^* \mathbf{f}_t = \sum_{i=1}^n k(\mathbf{x}_t, \mathbf{x}_i, \alpha_i)_j$ holds by (2). Moreover, we can observe that for $k(\mathbf{x}_t, \mathbf{x}_i, \alpha_i)_j$ to have a significant value, two conditions must be satisfied: (a) $\alpha_{ij}$ should have a large value, and (b) $\mathbf{f}_i^T \mathbf{f}_t$ should have a large value. Therefore, we interpret the pre-activation value $\Phi(\mathbf{x}_t, \boldsymbol{\Theta})_j$ as a weighted sum for the feature similarity $\mathbf{f}_i^T \mathbf{f}_t$ with the weight $\alpha_{ij}$. When $\mathbf{f}_t$ is close to $\mathbf{f}_i$ with a large positive weight $\alpha_{ij}$, the prediction score for class $j$ is increased. On the other hand, when $\mathbf{f}_t$ is close to $\mathbf{f}_i$ with a large negative weight $\alpha_{ij}$, the prediction score for class $j$ is then decreased.

We can thus interpret the training points with negative representer values as inhibitory points that suppress the activation value, and those with positive representer values as excitatory examples that does the opposite. We demonstrate this notion with examples further in Section 4.2. We note that such excitatory and inhibitory points provide a richer understanding of the behavior of the neural network: it provides insight both as to why the neural network prefers a particular prediction, as well as *why it does not*, which is typically difficult to obtain via other sample-based explanations.

## 3.1 Training an Interpretable Model by Imposing L2 Regularization.

Theorem 3.1 works for any model that performs a linear matrix multiplication before the activation $\sigma$, which is quite general and can be applied on most neural-network-like structures. By simply introducing a L2 regularizer on the weight with a fixed $\lambda > 0$, we can easily decompose the pre-softmax prediction value as some finite linear combinations of a function between the test and train data. We now state our main algorithm. First we solve the following optimization problem:

$$\boldsymbol{\Theta}^* = \arg\min_{\boldsymbol{\Theta}} \frac{1}{n} \sum_i^n L(\mathbf{y}_i, \Phi(\mathbf{x}_i, \boldsymbol{\Theta})) + \lambda ||\boldsymbol{\Theta}_1||^2. \tag{3}$$

Note that for the representer point selection to work, we would need to achieve a stationary point with high precision. In practice, we find that using a gradient descent solver with line search or LBFGS solver to fine-tune after converging in SGD can achieve highly accurate stationary point. Note that we can perform the fine-tuning step only on $\boldsymbol{\Theta}_1$, which is usually efficient to compute. We can then decompose $\Phi(\mathbf{x}_t, \boldsymbol{\Theta}) = \sum_i^n k(\mathbf{x}_t, \mathbf{x}_i, \alpha_i)$ by Theorem 3.1 for any arbitrary test point $\mathbf{x}_t$, where $k(\mathbf{x}_t, \mathbf{x}_i, \alpha_i)$ is the contribution of training point $\mathbf{x}_i$ on the pre-softmax prediction $\Phi(\mathbf{x}_t, \boldsymbol{\Theta})$. We emphasize that imposing L2 weight decay is a common practice to avoid overfitting for deep neural networks, which does not sacrifice accuracy while achieving a more interpretable model.

## 3.2 Generating Representer Points for a Given Pre-trained Model.

We are also interested in finding representer points for a given model $\Phi(\boldsymbol{\Theta}_{given})$ that has already been trained, potentially without imposing the L2 regularizer. While it is possible to add the L2 regularizer and retrain the model, the retrained model may converge to a different stationary point, and behave differently compared to the given model, in which case we cannot use the resulting representer points as explanations. Accordingly, we learn the parameters $\boldsymbol{\Theta}$ while imposing the L2 regularizer, but under the additional constraint that $\Phi(\mathbf{x}_i, \boldsymbol{\Theta})$ be close to $\Phi(\mathbf{x}_i, \boldsymbol{\Theta}_{given})$. In this case, our learning objective becomes $\Phi(\mathbf{x}_i, \boldsymbol{\Theta}_{given})$ instead of $y_i$, and our loss $L(\mathbf{x}_i, y_i, \boldsymbol{\Theta})$ can be written as $L(\Phi(\mathbf{x}_i, \boldsymbol{\Theta}_{given}), \Phi(\mathbf{x}_i, \boldsymbol{\Theta}))$.

**Definition 3.1.** We say that a convex loss function $L(\Phi(\mathbf{x}_i, \boldsymbol{\Theta}_{given}), \Phi(\mathbf{x}_i, \boldsymbol{\Theta}))$ is "suitable" to an activation function $\sigma$, if it holds that for any $\boldsymbol{\Theta}^* \in \arg\min_{\boldsymbol{\Theta}} L(\Phi(\mathbf{x}_i, \boldsymbol{\Theta}_{given}), \Phi(\mathbf{x}_i, \boldsymbol{\Theta}))$, we have $\sigma(\Phi(\mathbf{x}_i, \boldsymbol{\Theta}^*)) = \sigma(\Phi(\mathbf{x}_i, \boldsymbol{\Theta}_{given}))$.

Assume that we are given such a loss function $L$ that is "suitable to" the activation function $\sigma$. We can then solve the following optimization problem:

$$\boldsymbol{\Theta}^* \in \arg\min_{\boldsymbol{\Theta}} \left\{ \frac{1}{n} \sum_i^n L(\Phi(\mathbf{x}_i, \boldsymbol{\Theta}_{given}), \Phi(\mathbf{x}_i, \boldsymbol{\Theta})) + \lambda ||\boldsymbol{\Theta}_1||^2 \right\}. \tag{4}$$

The optimization problem can be seen to be convex under the assumptions on the loss function. The parameter $\lambda > 0$ controls the trade-off between the closeness of $\sigma(\Phi(\mathbf{X}, \boldsymbol{\Theta}))$ and $\sigma(\Phi(\mathbf{X}, \boldsymbol{\Theta}_{given}))$, and the computational cost. For a small $\lambda$, $\sigma(\Phi(\mathbf{X}, \boldsymbol{\Theta}))$ could be arbitrarily close to $\sigma(\Phi(\mathbf{X}, \boldsymbol{\Theta}_{given}))$, while the convergence time may be long. We note that the learning task in Eq. (4) can be seen as learning from a teacher network $\boldsymbol{\Theta}_{given}$ and imposing a regularizer to make the student model $\boldsymbol{\Theta}$ capable of generating representer points. In practice, we may take $\boldsymbol{\Theta}_{given}$ as an initialization for $\boldsymbol{\Theta}$ and perform a simple line-search gradient descent with respect to $\boldsymbol{\Theta}_1$ in (4). In our experiments, we discover that the training for (4) can converge to a stationary point in a short period of time, as demonstrated in Section 4.5.

We now discuss our design for the loss function that is mentioned in (4). When $\sigma$ is the soft-max activation, we choose the softmax cross-entropy loss, which computes the cross entropy between $\sigma(\Phi(\mathbf{x}_i, \boldsymbol{\Theta}_{given}))$ and $\sigma(\Phi(\mathbf{x}_i, \boldsymbol{\Theta}))$ for $L_{\text{softmax}}(\Phi(\mathbf{x}_i, \boldsymbol{\Theta}_{given}), \Phi(\mathbf{x}_i, \boldsymbol{\Theta}))$. When $\sigma$ is ReLU activation, we choose $L_{\text{ReLU}}(\Phi(\mathbf{x}_i, \boldsymbol{\Theta}_{given}), \Phi(\mathbf{x}_i, \boldsymbol{\Theta})) = \frac{1}{2} \max(\Phi(\mathbf{x}_i, \boldsymbol{\Theta}), 0) \odot \Phi(\mathbf{x}_i, \boldsymbol{\Theta}) - \max(\Phi(\mathbf{x}_i, \boldsymbol{\Theta}_{given}), 0) \odot \Phi(\mathbf{x}_i, \boldsymbol{\Theta})$, where $\odot$ is the element-wise product. In the following Proposition, we show that $L_{\text{softmax}}$ and $L_{\text{ReLU}}$ are convex, and satisfy the desired suitability property in Definition 3.1. The proof is provided in the supplementary material.

**Proposition 3.1.** *The loss functions $L_{softmax}$ and $L_{ReLU}$ are both convex in $\boldsymbol{\Theta}_1$. Moreover, $L_{softmax}$ is "suitable to" the softmax activation, and $L_{ReLU}$ is "suitable to" the ReLU activation, following Definition 3.1.*

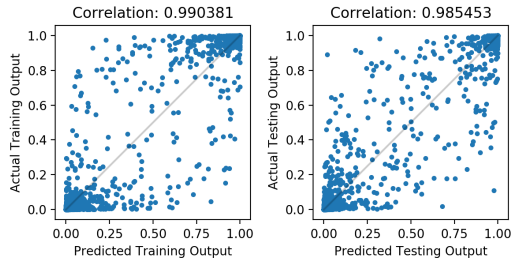

Figure 1: Pearson correlation between the actual and approximated softmax output (expressed as a linear combination) for train (left) and test (right) data in CIFAR-10 dataset. The correlation is almost 1 for both cases.

As a sanity check, we perform experiments on the CIFAR-10 dataset [15] with a pre-trained VGG-16 network [16]. We first solve (4) with loss $L_{\text{softmax}}(\Phi(\mathbf{x}_i, \Theta), \Phi(\mathbf{x}_i, \Theta_{given}))$ for $\lambda = 0.001$, and then calculate $\Phi(\mathbf{x}_t, \Theta^*) = \sum_{i=1}^{n} k(\mathbf{x}_t, \mathbf{x}_i, \alpha_i)$ as in (2) for all train and test points. We note that the computation time for the whole procedure only takes less than a minute, given the pre-trained model. We compute the Pearson correlation coefficient between the actual output $\sigma(\Phi(\mathbf{x}_t, \Theta))$ and the predicted output $\sigma(\sum_{i=1}^{n} k(\mathbf{x}_t, \mathbf{x}_i, \alpha_i))$ for multiple points and plot them in Figure 1. The correlation is almost 1 for both train and test data, and most points lie at the both ends of $y = x$ line.

We note that Theorem 3.1 can be applied to any hidden layer with ReLU activation by defining a sub-network from input $\mathbf{x}$ and the output being the hidden layer of interest. The training could be done in a similar fashion by replacing $L_{\text{softmax}}$ with $L_{\text{ReLU}}$. In general, any activation can be used with a derived "suitable loss".

## 4 Experiments

We perform a number of experiments with multiple datasets and evaluate our method's performance and compare with that of the influence functions.[2] The goal of these experiments is to demonstrate that selecting the representer points is efficient and insightful in several ways. Additional experiments discussing the differences between our method and the influence function are included in the supplementary material.

### 4.1 Dataset Debugging

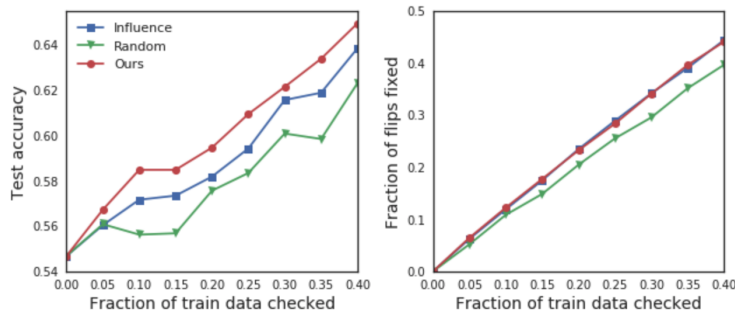

Figure 2: Dataset debugging performance for several methods. By inspecting the training points using the representer value, we are able to recover the same amount of mislabeled training points as the influence function (right) with the highest test accuracy compared to other methods (left).

To evaluate the influence of the samples, we consider a scenario where humans need to inspect the dataset quality to ensure an improvement of the model's performance in the test data. Real-world data is bound to be noisy, and the bigger the dataset becomes, the more difficult it will be for humans to look for and fix mislabeled data points. It is crucial to know which data points are more important than the others to the model so that prioritizing the inspection can facilitate the debugging process.

To show how well our method does in dataset debugging, we run a simulated experiment on CIFAR-10 dataset [17] with a task of binary classification with logistic regression for the classes automobiles and horses. The dataset is initially corrupted, where 40 percent of the data has the labels flipped, which naturally results in a low test accuracy of $0.55$. The simulated user will check some fraction of the train data based on the order set by several metrics including ours, and fix the labels. With the corrected version of the dataset, we retrain the model and record the test accuracies for each metrics. For our method, we train an explainable model by mimimizing (3) as explained in section 3.1. The L2 weight decay is set to $1e^{-2}$ for all methods for fair comparison. All experiments are repeated for 5 random splits and we report the average result. In Figure 2 we report the results for four different metrics: "ours" picks the points with bigger $|\alpha_{ij}|$ for training instance $i$ and its corresponding label $j$; "influence" prioritizes the training points with bigger influence function value; and "random" picks random points. We observe that our method recovers the same amount of training data as the influence function while achieving higher testing accuracy. Nevertheless, both methods perform better than the random selection method.

## 4.2 Excitatory (Positive) and Inhibitory (Negative) Examples

We visualize the training points with high representer values (both positive and negative) for some test points in Animals with Attributes (AwA) dataset [18] and compare the results with those of the influence functions. We use a pre-trained Resnet-50 [19] model and fine-tune on the AwA dataset to reach over 90 percent testing accuracy. We then generate representer points as described in section 3.2. For computing the influence functions, just as described in [10], we froze all top layers of the model and trained the last layer. We report top three points for two test points in the following Figures 3 and 4. In Figure 3, which is an image of three grizzly bears, our method correctly returns three images that are in the same class with similar looks, similar to the results from the influence function. The positive examples excite the activation values for a particular class and supports the decision the model is making. For the negative examples, just like the influence functions, our method returns images that look like the test image but are labeled as a different class. In Figure 4, for the image of a rhino the influence function could not recover useful training points, while ours does, including the similar-looking elephants or zebras which might be confused as rhinos, as negatives. The negative examples work as inhibitory examples for the model – they suppress the activation values for a particular class of a given test point because they are in a different class despite their striking similarity to the test image. Such inhibitory points thus provide a richer understanding, even to machine learning experts, of the behavior of deep neural networks, since they explicitly indicate training points that lead the network away from a particular label for the given test point. More examples can be found in the supplementary material.

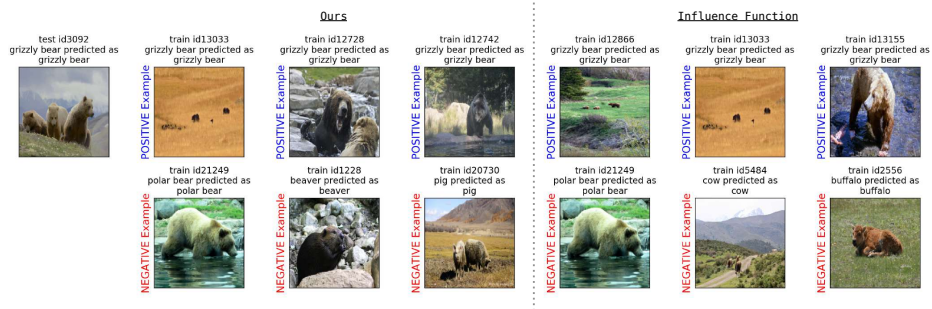

Figure 3: Comparison of top three positive and negative influential training images for a test point (left-most column) using our method (left columns) and influence functions (right columns).

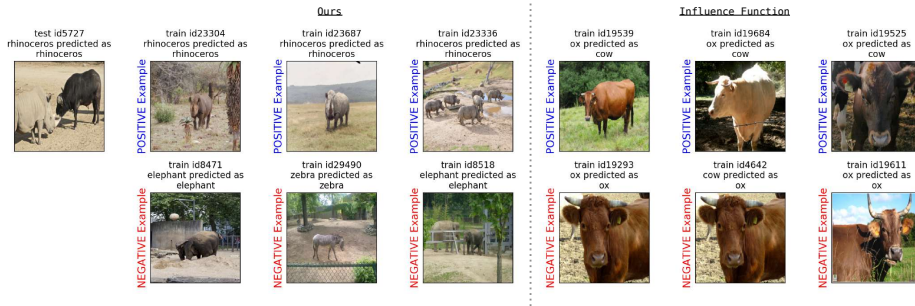

Figure 4: Here we can observe that our method provides clearer positive and negative examples while the influence function fails to do so.

## 4.3 Understanding Misclassified Examples

The representer values can be used to understand the model's mistake on a test image. Consider a test image of an antelope predicted as a deer in the left-most panel of Figure 5. Among 181 test images of antelopes, the total number of misclassified instances is 15, among which 12 are misclassified as deer. All of those 12 test images of antelopes had the four training images shown in Figure 5 among the top inhibitory examples. Notice that we can spot antelopes even in the images labeled as zebra or elephant. Such noise in the labels of the training data confuses the model – while the model sees elephant *and* antelope, the label forces the model to focus on just the elephant. The model thus learns to inhibit the antelope class given an image with small antelopes and other large objects. This insight suggests for instance that we use multi-label prediction to train the network, or perhaps clean the dataset to remove such training examples that would be confusing to humans as well. Interestingly, the model makes the same mistake (predicting deer instead of antelope) on the second training image shown (third from the left of Figure 5), and this suggests that for the training points, we should expect most of the misclassifications to be deer as well. And indeed, among 863 training images of antelopes, 8 are misclassified, and among them 6 are misclassified as deer.

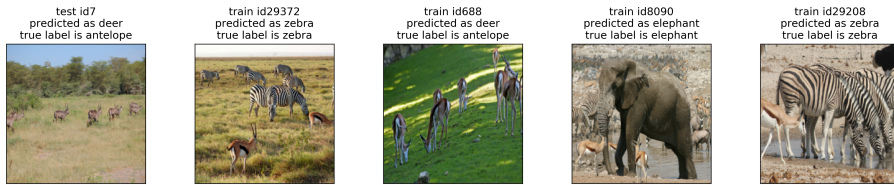

Figure 5: A misclassified test image (left) and the set of four training images that had the most negative representer values for almost all test images in which the model made the same mistakes. The negative influential images all have antelopes in the image despite the label being a different animal.

## 4.4 Sensitivity Map Decomposition

From Theorem 3.1, we have seen that the pre-softmax output of the neural network can be decomposed as the weighted sum of the product of the training point feature and the test point feature, or $\Phi(\mathbf{x}_t, \boldsymbol{\Theta}^*) = \sum_i^n \alpha_i \mathbf{f}_i^T \mathbf{f}_t$. If we take the gradient with respect to the test input $\mathbf{x}_t$ for both sides, we get $\frac{\partial \Phi(\mathbf{x}_t, \boldsymbol{\Theta}^*)}{\partial \mathbf{x}_t} = \sum_i^n \alpha_i \frac{\partial \mathbf{f}_i^T \mathbf{f}_t}{\partial \mathbf{x}_t}$. Notice that the LHS is the widely-used notion of sensitivity map (gradient-based attribution), and the RHS suggests that we can decompose this sensitivity map into a weighted sum of sensitivity maps that are native to each $i$-th training point. This gives us insight into how sensitivities of training points contribute to the sensitivity of the given test image.

In Figure 6, we demonstrate two such examples, one from the class zebra and one from the class moose from the AwA dataset. The first column shows the test images whose sensitivity maps we wish to decompose. For each example, in the following columns we show top four influential representer

points in the the top row, and visualize the decomposed sensitivity maps in the bottom. We used SmoothGrad [20] to obtain the sensitivity maps.

For the first example of a zebra, the sensitivity map on the test image mainly focuses on the face of the zebra. This means that infinitesimally changing the pixels around the face of the zebra would cause the greatest change in the neuron output. Notice that the focus on the head of the zebra is distinctively the strongest in the fourth representer point (last column) when the training image manifests clearer facial features compared to other training points. For the rest of the training images that are less demonstrative of the facial features, the decomposed sensitivity maps accordingly show relatively higher focus on the background than on the face. For the second example of a moose, a similar trend can be observed – when the training image exhibits more distinctive bodily features of the moose than the background (first, second, third representer points), the decomposed sensitivity map highlights the portion of the moose on the test image more compared to training images with more features of the background (last representer point). This provides critical insight into the contribution of the representer points towards the neuron output that might not be obvious just from looking at the images itself.

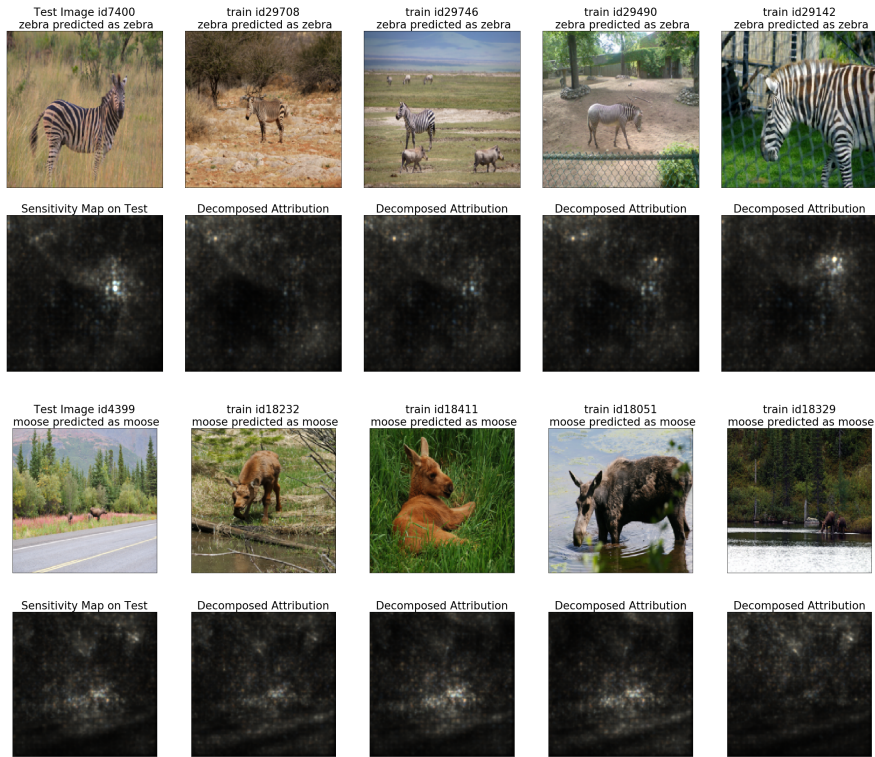

Figure 6: Sensitivity map decomposition using representer points, for the class zebra (above two rows) and moose (bottom two rows). The sensitivity map on the test image in the first column can be readily seen as the weighted sum of the sensitivity maps for each training point. The less the training point displays spurious features from the background and more of the features related to the object of interest, the more focused the decomposed sensitivity map corresponding to the training point is at the region the test sensitivity map mainly focuses on.

## 4.5   Computational Cost and Numerical Instabilities

Computation time is particularly an issue for computing the influence function values [10] for a large dataset, which is very costly to compute for each test point. We randomly selected a subset of test points, and report the comparison of the computation time in Table 1 measured on CIFAR-10 and AwA datasets. We randomly select 50 test points to compute the values for all train data, and recorded the average and standard deviation of computation time. Note that the influence function does not need the fine-tuning step when given a pre-trained model, hence the values being 0, while our method

first optimizes for $\Theta^*$ using line-search then computes the representer values. However, note that the fine-tuning step is a one time cost, while the computation time is spent for every testing image we analyze. Our method significantly outperforms the influence function, and such advantage will favor our method when a larger number of data points is involved. In particular, our approach could be used for *real-time explanations* of test points, which might be difficult with the influence function approach.

| | Influence Function | | Ours | |
|---|---|---|---|---|
| Dataset | Fine-tuning | Computation | Fine-tuning | Computation |
| CIFAR-10 | 0 | $267.08 \pm 248.20$ | $7.09 \pm 0.76$ | $0.10 \pm 0.08$ |
| AwA | 0 | $172.71 \pm 32.63$ | $12.41 \pm 2.37$ | $0.19 \pm 0.12$ |

Table 1: Time required for computing an influence function / representer value for all training points and a test point in seconds. The computation of Hessian Vector Products for influence function alone took longer than our combined computation time.

While ranking the training points according to their influence function values, we have observed numerical instabilities, more discussed in the supplementary material. For CIFAR-10, over 30 percent of the test images had all zero training point influences, so influence function was unable to provide positive or negative influential examples. The distribution of the values is demonstrated in Figure 7, where we plot the histogram of the maximum of the absolute values for each test point in CIFAR-10. Notice that over 300 testing points out of 1,000 lie in the first bin for the influence functions (right). We checked that all data in the first bin had the exact value of 0. Roughly more than 200 points lie in range $[10^{-40}, 10^{-28}]$, the values which may create numerical instabilities in computations. On the other hand, our method (left) returns non-trivial and more numerically stable values across all test points.

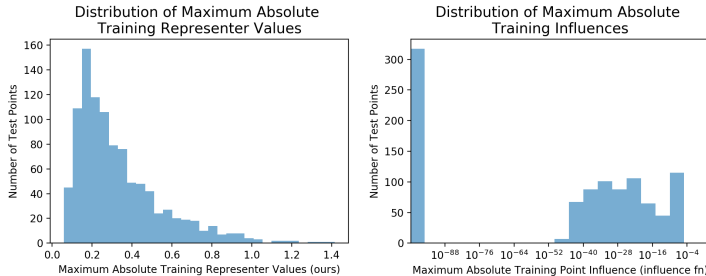

Figure 7: The distribution of influence/representer values for a set of randomly selected 1,000 test points in CIFAR-10. While ours have more evenly spread out larger values across different test points (left), the influence function values can be either really small or become zero for some points, as seen in the left-most bin (right).

## 5 Conclusion and Discussion

In this work we proposed a novel method of selecting representer points, the training examples that are influential to the model's prediction. To do so we introduced the modified representer theorem that could be generalized to most deep neural networks, which allows us to linearly decompose the prediction (activation) value into a sum of representer values. The optimization procedure for learning these representer values is tractable and efficient, especially when compared against the influence functions proposed in [10]. We have demonstrated our method's advantages and performances on several large-scale models and image datasets, along with some insights on how these values allow the users to understand the behaviors of the model.

An interesting direction to take from here would be to use the representer values for data poisoning just like in [10]. Also to truly see if our method is applicable to several domains other than image dataset with different types of neural networks, we plan to extend our method to NLP datasets with recurrent neural networks. The result of a preliminary experiment is included in the supplementary material.

**Acknowledgements**

We acknowledge the support of DARPA via FA87501720152, and Zest Finance.

## Footnotes

[2]Source code available at `github.com/chihkuanyeh/Representer_Point_Selection`.

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
