[Supplementary Material]

# A  Proof of Proposition 3.1

*Proof.* The convexity can be checked easily. If $\mathbf{\Theta}^* \in \arg\min_{\mathbf{\Theta}} L_{\text{softmax}}(\Phi(\mathbf{x}_i, \mathbf{\Theta}_{given}), \Phi(\mathbf{x}_i, \mathbf{\Theta}))$, by the first order condition we have

$$\frac{\partial L_{\text{softmax}}(\Phi(\mathbf{x}_i, \mathbf{\Theta}_{given}), \Phi(\mathbf{x}_i, \mathbf{\Theta}^*))}{\partial \mathbf{\Theta}_1^*} = \mathbf{0} \tag{5}$$

By chain rule we obtain

$$\frac{\partial L_{\text{softmax}}(\Phi(\mathbf{x}_i, \mathbf{\Theta}_{given}), \Phi(\mathbf{x}_i, \mathbf{\Theta}^*))}{\partial \Phi(\mathbf{x}_i, \mathbf{\Theta}^*)} \frac{\partial \Phi(\mathbf{x}_i, \mathbf{\Theta}^*)}{\partial \mathbf{\Theta}_1^*} = \mathbf{0}, \tag{6}$$

and thus we have

$$\left( \frac{\partial L_{\text{softmax}}(\Phi(\mathbf{x}_i, \mathbf{\Theta}_{given}), \Phi(\mathbf{x}_i, \mathbf{\Theta}^*))}{\partial \Phi(\mathbf{x}_i, \mathbf{\Theta}^*)} \right) \mathbf{f}_i^T = \mathbf{0}, \tag{7}$$

When $\mathbf{f}_i$ is not a zero vector, this reduces to

$$\frac{\partial L_{\text{softmax}}(\Phi(\mathbf{x}_i, \mathbf{\Theta}_{given}), \Phi(\mathbf{x}_i, \mathbf{\Theta}^*))}{\partial \Phi(\mathbf{x}_i, \mathbf{\Theta}^*)} = \mathbf{0}, \tag{8}$$

and we show that $\sigma(\Phi(\mathbf{x}_i, \mathbf{\Theta})) - \sigma(\Phi(\mathbf{x}_i, \mathbf{\Theta}_{given})) = \mathbf{0}$. As a result, $L_{\text{softmax}}$ is "suitable to" the softmax activation.

If $\mathbf{\Theta}^* \in \arg\min_{\mathbf{\Theta}} L_{\text{ReLU}}(\Phi(\mathbf{x}_i, \mathbf{\Theta}_{given}), \Phi(\mathbf{x}_i, \mathbf{\Theta}))$, we can rewrite $L_{\text{ReLU}}$ as

$$\begin{aligned} &L_{\text{ReLU}}(\Phi(\mathbf{x}_i, \mathbf{\Theta}_{given}), \Phi(\mathbf{x}_i, \mathbf{\Theta})) \\ &= \frac{1}{2} \max(\Phi(\mathbf{x}_i, \mathbf{\Theta}), 0) \odot \Phi(\mathbf{x}_i, \mathbf{\Theta}) - \Phi(\mathbf{x}_i, \mathbf{\Theta}_{given}) \odot \Phi(\mathbf{x}_i, \mathbf{\Theta}) \\ &= \sum_{j=0}^{c} \frac{1}{2} \max(\Phi_j(\mathbf{x}_i, \mathbf{\Theta}), 0) \Phi_j(\mathbf{x}_i, \mathbf{\Theta}) - \Phi_j(\mathbf{x}_i, \mathbf{\Theta}_{given}) \Phi_j(\mathbf{x}_i, \mathbf{\Theta}), \end{aligned} \tag{9}$$

and we note that $\mathbf{\Theta}_{1j}$, the $j$-th row for $\mathbf{\Theta}_1$, is only related to $\Phi_j(\mathbf{x}_i, \mathbf{\Theta})$. Therefore, we have $\mathbf{\Theta}_{1j}^* \in \arg\min_{\mathbf{\Theta}_{1j}} L_{\text{ReLU}}(\Phi_j(\mathbf{x}_i, \mathbf{\Theta}_{given}), \Phi_j(\mathbf{x}_i, \mathbf{\Theta}))$. We now consider the cases where $\max(\Phi_j(\mathbf{x}_i, \mathbf{\Theta}_{given}), 0) = 0$ and $\max(\Phi_j(\mathbf{x}_i, \mathbf{\Theta}_{given}), 0) > 0$.

When $\max(\Phi_j(\mathbf{x}_i, \mathbf{\Theta}_{given}), 0) = 0$,

$$L_{\text{ReLU}}(\Phi_j(\mathbf{x}_i, \mathbf{\Theta}_{given}), \Phi_j(\mathbf{x}_i, \mathbf{\Theta})) = \frac{1}{2} \max(\Phi_j(\mathbf{x}_i, \mathbf{\Theta}), 0) \cdot \Phi_j(\mathbf{x}_i, \mathbf{\Theta}). \tag{10}$$

$\mathbf{\Theta}_{1j}$ obtains the minimum when $\max(\Phi_j(\mathbf{x}_i, \mathbf{\Theta}), 0) = 0$, therefore $\sigma(\Phi_j(\mathbf{x}_i, \mathbf{\Theta})) = \sigma(\Phi_j(\mathbf{x}_i, \mathbf{\Theta}_{given}))$.

When $\max(\Phi_j(\mathbf{x}_i, \mathbf{\Theta}_{given}), 0) > 0$,

$$L_{\text{ReLU}}(\Phi_j(\mathbf{x}_i, \mathbf{\Theta}_{given}), \Phi_j(\mathbf{x}_i, \mathbf{\Theta})) = \frac{1}{2} \max(\Phi_j(\mathbf{x}_i, \mathbf{\Theta}), 0) \cdot \Phi_j(\mathbf{x}_i, \mathbf{\Theta}) - \Phi_j(\mathbf{x}_i, \mathbf{\Theta}_{given}) \cdot \Phi_j(\mathbf{x}_i, \mathbf{\Theta}).$$

For $\Phi_j(\mathbf{x}_i, \mathbf{\Theta}) \leq 0$, the minimum for $L_{\text{ReLU}}$ is 0. For $\Phi_j(\mathbf{x}_i, \mathbf{\Theta}) > 0$, the minimum for $L_{\text{ReLU}}$ is $-\frac{1}{2}\Phi_j(\mathbf{x}_i, \mathbf{\Theta}_{given})^2$ only if $\Phi_j(\mathbf{x}_i, \mathbf{\Theta}) = \Phi_j(\mathbf{x}_i, \mathbf{\Theta}_{given})$. Therefore, the minimum is reached again when $\sigma(\Phi_j(\mathbf{x}_i, \mathbf{\Theta})) = \sigma(\Phi_j(\mathbf{x}_i, \mathbf{\Theta}_{given}))$. As a result, $L_{\text{ReLU}}$ is "suitable to" the ReLU activation. $\qquad\square$

# B  Relationship with the Influence Function

In this section, we compare the behaviors of our method and the influence functions [10]. Recall that for a training point $\mathbf{x}_i$ and a test point $\mathbf{x}_t$, the influence function value is computed in terms of the gradients/hessians of the loss, and it reflects how the loss at a test point $\mathbf{x}_t$ will change when the training point $\mathbf{x}_i$ is perturbed (weighted more/less). Because the influence function is defined in

Figure 8: Demonstration of numerical stability on a 2-D toy data. Even with a big margin, our values faithfully provide positive examples from the same class and negative examples from the different class near the decision boundary. Influence functions return zeros for all training points, thus is not able to provide influential points.

Figure 9: Euclidean vs Influence Function vs Representer Value (ours). Representer values have bigger scale in general. Some examples from regions (a) where the influence function value is zero while our representer value is big and (b) where our representer value is small while the influence function value is big are shown.

terms of the loss, its value can easily become arbitrarily close to zero if the the loss is flat in some region. On the other hand, our representer values are computed using the neurons' activation values, which may result in comparatively larger values in general. We verify this in a toy dataset in 2-D with a large margin shown in Figure 8.

We train a multi-layer perceptron with ReLU activations as a binary classifier. The influential points, we would expect, are the points that are closer to the decision boundary. As shown on the left of Figure 8, the influence function does not provide positive or negative examples from each class for the given test point in green square because all training points have exactly zero influence function values, marked with cyan crosses. However, our method provides correct positive and negative points near the decision boundary as we can see from the rightmost panel of Figure 8.

In Figure 9, we compare the behaviors of several metrics (Euclidean distance, influence function, representer value) for selecting training points that are most similar to a test point from CIFAR-10 dataset. We use a pre-trained VGG-16. As we can observe from the first two plots, the Euclidean distance does not reflect the class each training point is in – even if the training point is far away from the test point it may still be a similar image in the same class. On the other hand, representer and influence function values tell us about which class each training point is in. From the third plot, we observe that influence function and representer values agree on selecting images of horses as harmful and dogs as helpful.

## C    More Examples of Positive/Negative Reperesenter Points

More examples of positive and negative representer points are shown in Figure 10. Observe that the positive points all have the same class label as the test image with high resemblance, while the negative points, despite their similarity to the test image, have different classes.

Figure 10: More examples of positive and negative represener points. The first column is composed of test images from different classes from AwA dataset; the next three images are the positive representer points; the next three are negative representer points for each test image.

# D    Representer Points of LSTM on NLP Data

We perform a preliminary experiment on an LSTM network trained on a IMDB movie review dataset [21]. Each data point is a review about a movie, and the task is to identify whether the review has a positive or negative sentiment. The pretrain model achieves $87.5\%$ accuracy. We obtain the positive and negative repesenter points with methods described in section 3.2. In Table 2, we show the test review which is predicted as a negative review by the LSTM network. We observe that both the top-1 positive and negative representer contain negative connotations just like the test review, but in the negative representer (which is a positive review about the movie) the negative connotation comes from a character in the movie rather than from a reviewer.

| | |
|---|---|
| Test Review | \<START\> when i first saw this movie in the theater i was so angry it completely blew in my opinion i didn't see it for a decade then decided what the hell let's see i'm watching all <> movies now to see where it went wrong my guess is it was with sequel 5 that was the first to <> the whole i am in a dream <> i see weird stuff oh <> it's not a dream oh wait i see something spooky oh never mind <> storyline those which made it so scary in the first place nothing fantasy nothing weird the box got opened boom they came was the only one that could bargain her way out of it first |
| Positive Representer | \<START\> no not the <> of <> the <> <> <> but the mini series <> <> lifetime must have realized what a dog this was because the series was burned off two episodes at a time most of them broadcast between 11 p m friday nights and 1 a m saturday <> as to why i watched the whole thing i can only <> to <> sudden <> attacks of <> br br most of the cast are <> who are likely to remain unknown the only two <> names are shirley jones and rachel ward who turn in the only decent performances jones doesn't make it through the entire series lucky woman ward by the way is aging quite well since her |
| Negative Representer | \<START\> this time around <> is no longer royal or even particularly close to being any such thing instead rather a butler to the prince <> portrayed by hugh <> who <> tim <> who presence is <> missed and that hole is never filled his character had an innocent charm was a bumbling and complete <> we can't help but care for him which isn't at all true of his <> as being <> which he apparently was according to the <> page not to mention loud <> and utterly non threatening <> can now do just about what he <> and does so why is he so frustrated and angry honestly it gets depressing at times yes his master is a <> they |

Table 2: An example of top-1 positive and negative representer points for IMDB dataset. <> stands for unknown words since only top 5000 vocabularies are used.