[Reviews · NeurIPS 2018]

Reviewer 1



This paper proposes a decomposition of the pre-activation prediction (the values of the last intermediate layer in a NN) into a linear combination of activations of the training points. The weights in this linear combination are called representer values. Positive representer values represent excitatory signals that contribute to the prediction of the particular sample in the corresponding class, while negative representer values inhibit the prediction to that particular class. The representer values can be used to better understand the prediction of the model. The experimental section shows how this technique can be used in several explanatory analyzes, such as: * Data debugging: for a MNIST dataset, consider some of the labels being interchanged in the dataset (in the example used some 1s become 7s). With this corrupted data, the binary prediction (distinguishing 1s from 7s) has a low accuracy of 85%. Using the representer values, a simulated user checks a fraction of the labels of the dataset and flips back the labels accordingly. The experimental results shows that when using representer values to guide the flipping of the label, the accuracy of the classification recovers faster (i.e., a smaller fraction of the data being checked leads to higher classification) than when using other methods for guiding the flipping (loss-based, random and influence functions). * Positive and negative examples: This section compares the positive and negative examples given by the proposed method and the influence functions. The proposed method seems more consistent in providing the right type of examples. * Understanding misclassified examples: For a few misclassified examples of antelopes, the representer values are used to inspect the training samples that inhibit the correct prediction. An interesting discovery is made: the highly inhibitory training points contain an antelope together with other (larger) animals and the true label associated with these points is the other animal. Overall I really enjoyed reading this paper, specially the experimental section. The technique introduced seems to be useful in all kind of debugging experiments. My favorite one was the understanding of misclassified examples. In addition, it seems to perform slightly better than influence functions and it's significantly faster to compute. I am curious how this method could be extended to tasks that are not classification tasks. For example, to NLP tasks that detect spans in a sequence of text. Also, is it feasible to include the computation of the representer values in some type of library or within some of the popular frameworks such that it becomes a tool readily available? Nit-picks: * Are citations 13 and 14 swapped? * Line 91: do you mean a generic yi or only the yi the predicted label of xi (I think it's the former and maybe use a different index for y to denote that it's a generic - let's say - yt) UPDATE: After rebuttal and discussion, I'm lowering my score. Some more feedback: Quantitative results in Fig 3 When I looked again at the results presented in Fig 3, I'm not sure I understand the correlation between the two graphs in that figure. Let's look at 0.15 on x axis. Loss (green) and Influence (blue) manage to flip a higher percentage of the mislabeled samples than random (red) (graph on the right); however, retraining leads to higher accuracy for random (red) than loss (green) and influence (blue) (left graph). Counterintuitive to me. What am I missing? In particular, random being better than influence is counterintuitive. In general, as a reader, I expect the authors to discuss the results obtained, not only stating: here it is, ours is better. I was enthusiastic when reading this paper as I think we need more work in AI that addresses explainability, model debugging and data set debugging. This paper introduces such a method. After reading the other reviewers' comments (in particular R2), I agree that the interpretation of the negative vs. positive examples is subjective and neither the paper nor the rebuttal manage to quantify the importance of the positive/negative representers. I was thinking about the following experiment: pick a test sample (predicted correctly by the model), sort representer points (both negative and positive), sort them on absolute value, and start removing the top 5%, 10%, ... retrain the model and see when the test sample is misclassified. This would be an indirect measure for the importance of the representer points in correctly classifying the test sample.

Reviewer 2



The manuscript proposes a method to measure the effect that training examples have on the performance of a deep neural network (DNN). Towards this goal, a method is proposed to identify "representer" points that measure the importance that a training image has on the performance of a given model. The paper proposes to identify representer points of two types: i) positive points; which correspond to examples that support the decisions made by the network, and ii) negative points; which inhibit the believe of the model towards certain predictions. Experiments with the VGG-16 and ResNet-50 architectures on the MNIST, CIFAR-10, and Animals with Attributes (AWA) datasets show the strenghts of the proposed method. The proposed method is sound and well motivated. A detailed formal presentation of the method is provided. In addition, aspects related to the computational cost and numerical properties of the proposed method are adequately discussed. I find the dataset debugging experiment (Sec.4.1) an interesting means to asses the effect of training examples on the learning of the model being trained. My main concerns with the manuscript are the following: i) I consider there is some room for improvement regarding the formal presentation of the method since it can be a bit complex to go through it at the begining. For instance, initially I had troubles understanding what the text "the last intermediate layer feature in the neural network" in l.82 refers to. ii) While positive/negative representers could be good indicators of the performance of the model, in general, I find hard to find it as a good explanatory means for a specific given test example. For instance, in Fig.4 images of influential examples are presented. However, it is difficult for me to make the direct connection on how these examples "explain" the prediction made for the test image. In my opinion the debugging experiment (Sec.4.1), combined with its related discussion in Sec.4.3 seem to be a more effective use of the proposed method, and perhaps a potentially stronger contribution, than that of prediction explainability that is claimed by the paper. iii) Finally, the presented results (Fig. 4,5,6) for the model explanations are mostly qualitative which can be to some extent subjective. In this regard, I find the amount of provided qualitative examples quite reduced. Therefore, I encourage the authors to add supplementary material in order to show extended qualitative results of the explanations produced by their method. Perhaps this can further demonstrate the explanation potential of the proposed method. I would appreciate if my concerns are addressed in the rebuttal. ======================== Post rebuttal update ======================== Having read the reviews and the provided rebuttal, my impression of the manuscript is the following. I do appreciate that a significant part of the rebuttal is focused on addressing my comments. However, I am still not convinced regarding the usefulness of the positive/negative representers as means for explanation. At the end of Sec.1 there is an example highlighting the un-annotated class instances, e.g. antilopes, as negative representers. Later in Sec. 4.2 more examples where the negative representers are images with instances of other classes with similar shape to that of the class of interest are discussed. Similarly, in the rebuttal an example regarding bears was provided. Still very similar to what was presented on the paper. I would be more convinced about the explanation capabilities of the method if, on top of these representers, there would be an overlaid visualization, e.g. a grad-CAM (Selvaraju et a., arXiv:1610.02391), CNN-Fixation (Reddy et al., arXiv:1708.06670 ), DeconvNet (Springenberg et al. ICLR'15), etc. heatmap, highlighting the visual features that make this image an actual negative representer, e.g. highlighting the un-annotated antilopes. At this point, it is the person visualizing the representer negative image, the one coming out with the explanation. Having said this, and as stated on my initial review, I do find quite valuable the use of these representer points as means for debugging (Sec. 4.1 and Sec 4.3). I share the opinion of the other reviewers that the proposed method holds good promise as a model debugging tool. Moreover, I motivate the authors to update the manuscript to present their method as a debugging method instead of as a tool for explanation. In this regard, I still feel the explanation capability of the proposed method not fully justified. Finally, I appreciate the additional qualitative examples. I will encourage the authors to enrich their manuscript by adding supplementary material with a good sample of these examples.

Reviewer 3



This paper presents a novel method for explaining black-box models. The model proposes to find representer points, i.e., the training examples that are influential to the model’s prediction and decompose the prediction to the sum of representer values. By learning the weights corresponding to the representer values, the importance of different training samples are indicated. Extensive experimental results are provided to show the validity of the method, such as data debugging, misclassification analysis, etc. The proposed method is valid and provide insightful interpretation on the results for black-box models. The experimental results seem promising. The presentation is clear and easy to follow. This is a good work on interpretation. I only have one concern. For interpretation of a black-box model, this method needs to know how the black-box model is formulated and seems not applicable when the loss function of the black-box model is not available. ------------------------------------------------------------------------------------ AFTER REBUTTAL: Having read all the reviews and the feedback from the authors, I would like to lower my score a bit (7->6) for the following reasons: 1. As I checked Section 4.1 again, I am a bit concerned about the validity of data debugging setting, as the authors propose to check data with larger |k(xi, xi, αi)| values. I am not sure if the representer values can be a good indicator of label correctness. Does higher training point influence necessarily suggest larger mislabel probability? In particular, training points with large positive representer values tend to be similar samples with correct labels as shown in Fig. 4 and 5. 2. I agree with other reviewers that we should not be limited by the desire on quantitative results in a research paper. However, since the authors propose a method for training an interpretable model in Section 3.1, it would be helpful to report the performance (e.g., prediction accuracy) of such interpretable model, at least on some toy data sets, like MNIST. 3. Compared with influence function, Table 1 provides a compelling advantage in terms of computational efficiency. This is the main reason for my enthusiasm about this work, as efficiency is critical for applying an interpretation method to large-scale scenarios. However, the comparison on explanation quality seems subjective. I could see that Fig. 5 provides some better negative example choices. While in Fig. 2, the definition of "dogness" seems a bit vague and subjective to me. In general, I think this paper presents an interesting idea (somewhat reminiscent of the layer-wise relevance propagation method, which is yet for features-based explanation). Compared with influence function, this method has been proven to be more scalable. The comparison on explanation quality is a bit subjective and may be improved via additional results.